# Validation of the English Version of the General Dietary Behavior Inventory (GDBI-E)

**DOI:** 10.3390/ijerph19052883

**Published:** 2022-03-01

**Authors:** Matthias Marsall, Gerrit Engelmann, Eva-Maria Skoda, Nanette Stroebele-Benschop, Martin Teufel, Alexander Bäuerle

**Affiliations:** 1Clinic of Psychosomatic Medicine and Psychotherapy, LVR-University Hospital, University of Duisburg-Essen, 45147 Essen, Germany; matthias.marsall@ukbonn.de (M.M.); gerrit.engelmann@lvr.de (G.E.); eva-maria.skoda@uni-due.de (E.-M.S.); martin.teufel@uni-due.de (M.T.); 2Institute for Patient Safety (IfPS), University Hospital Bonn, 53127 Bonn, Germany; 3Center for Translational Neuro- and Behavioral Sciences (C-TNBS), University of Duisburg-Essen, 45147 Essen, Germany; 4Institute of Nutritional Medicine, University Hohenheim, 70599 Hohenheim, Germany; n.stroebele@uni-hohenheim.de

**Keywords:** dietary behavior, assessment, dietary recommendation, body mass index, construct validation, criterion validity, cluster analysis

## Abstract

In medical science and practice, dietary behavior is mostly assessed by item-extensive questionnaires (e.g., food-frequency-questionnaires) or by questionnaires focusing on psychological aspects of dietary behavior neglecting dietary quality or quantity. In consequence, these questionnaires do not capture the full bandwidth of dietary behavior or are less effective in the assessment of dietary behavior because of the large item pools. Therefore, the aim of this validation study was to translate the existing General Dietary Behavior Inventory (GDBI), which was constructed as a behavior-related, as well as effective, instrument, and verifying its construct and criterion validity. This inventory is based on the general nutrition recommendations of the World Health Organization (WHO). Our English-speaking convenience sample consisted of 263 participants. The study results confirmed convergent, as well as criterion validity of the English version of the GDBI (GDBI-E). Discriminant validity of the GDBI-E could mainly be verified. Different dietary behavior clusters were identified in a cluster analysis. The found clusters represented a rather healthy and a rather unhealthy dietary behavior in the sample according to the recommendations of the WHO. The results underpinned the validity of the GDBI-E. The GDBI-E is applicable in research and clinical practice to assess dietary behavior in the English-speaking population.

## 1. Introduction

Human behaviors regarding one’s own health status have been identified in different studies to be relevant resources in maintaining or promoting health [1,2,3]. Especially, dietary behavior of individuals has been identified as a health-promoting resource [4]. Dietary behavior describes a wide range of behaviors occurring in the context of dietary intake [5]. It includes different behavioral options of individuals handling foods and their nutrition. Furthermore, dietary behavior is not only limited to food, energy or nutrient intake or psychological components, but also to concrete behaviors associated with dietary intake [6]. Thus, dietary behavior represents typical diet-related habits, which influence one’s own health status.

The assessment of dietary behavior is conducted in different ways in clinical research that has been described by Engelmann and colleagues [7]. The types of assessment range from food frequency questionnaires, (e.g., [8,9]), over psychological assessments, (e.g., [10,11]), or assessments of disorder-specific eating behaviors (e.g., [12,13]), to questionnaires for the qualitative assessment of dietary behavior (e.g., [14,15]). However, these questionnaire formats hardly capture concrete dietary behavior. Especially, food frequency questionnaires and questionnaires for the qualitative assessment of dietary behavior are not effective tools to assess dietary behavior due to their high number of items [8,9]. On the other hand, psychological assessments of dietary behavior do not consider dietary quality or quantity. Thus, it is evident that these assessment tools do not capture concrete dietary behavior, which is shown in daily or ordinary life situations representing people’s dietary decisions. The development of the General Dietary Behavior Inventory (GDBI) [7] takes this into account by assessing relevant aspects of dietary behavior going beyond mere food or nutrient intake. It looks at people’s food choices in life situations, as well as food handling, such as preparation of a meal and the way that food is eaten. The GDBI is based on theoretical and reliable recommendations of the World Health Organization (WHO) and the German Nutrition Society (DGE) implying relevant recommendations for the favorable food choices and a positive dietary behavior. 

In order to make the GDBI also accessible to the English-speaking population, the goal of this study was to conduct a content-related translation of the GDBI into the English language and to validate the English version. This is the first study to translate and validate the GDBI in another language. Therefore, convergent and discriminant validity were examined to confirm construct validity. Furthermore, criterion validity of the GDBI-E should be verified. To prove different dietary clusters according to international nutrition recommendations, a cluster analysis should be conducted.

## 2. Materials and Methods

### 2.1. Development of the General Dietary Behavior Inventory in the English Language

The “General Dietary Behavior Inventory” (GDBI) for the assessment of general dietary behavior has already been developed and validated for a German-speaking population [7]. This questionnaire is based on general nutrition recommendations of the WHO [16,17] and the DGE [18,19,20]. The existing 16-item questionnaire was translated from German language into English independently by two authors. Subsequently, both translations were compared and discussed to derive a preliminary final translation. Afterwards, an English native speaker crosschecked the preliminary final items and confirmed content equivalence with the original items. We specified these items as the final translated version and used the developed 5-point bipolar scale depicting concrete opposite dietary behaviors in terms of a semantic differential [21]. A sample item is “I eat at least 2 servings of fruit daily.” compared to “I never eat fruit.”. The scoring of each item is from 5 (=Like behavior A) to 1 (=Like behavior B), except for two inverted items (item codes db12(r) and db14(r)). The respective opposite behavior is rated with the score 1 while medium expressions receive scores from 2 to 4. The scores of each item (1 to 5) are added up to calculate the GDBI-E score. Higher GDBI-E scores represent a healthier dietary behavior regarding the dietary recommendations. The items and the scoring are shown in Appendix A. The originally developed German GDBI items were presented in the validation study by Engelmann and colleagues [7]. The GDBI-E measures dietary behavior in terms of a formative approach. This indicates that the individual items do not necessarily have to be correlated with each other, as they reflect different behaviors (e.g., a person may be likely to drink more than 1.5 L of fluid per day (item db11) but may also eat sweets daily (item db8). Therefore, a cluster analysis approach instead of factor analysis was performed in the validation of the GDBI-E. We assumed that different clusters should emerge representing a rather healthy and a rather less healthy dietary behavior.

### 2.2. Participants and Study Design

Participants for a cross-sectional online survey were recruited in international online networks (e.g., Facebook) and via personal and occupational networks. From a sample of 2493 people, 406 people declared their electronic informed consent to study participation; of this, 263 people participated in this study. In detail, we defined the following exclusion criteria: As sum scores had to be calculated for dietary behavior and nutrition knowledge, participants who did not complete any items of these constructs were excluded. In addition, participants who completed the survey in less than 6:30 min (0.05 quantile of the sample) or more than 25:58 min (0.95 quantile of the sample) were excluded. By excluding extremely fast and extremely slow responders, we ensured that our analysis was based on an average sample. Hereby, possible biases in response behavior were minimized. Moreover, we excluded one participant because of being younger than 18 years old and 11 participants because of obviously incorrect statements regarding their weight or height resulting in inappropriate calculation of body mass index (BMI). After these exclusions, the final sample was *n* = 263 participants reflecting a completion rate of 62% in relation to the total number of people who viewed the consent form. Average time for answering the survey was 12:08 min (SD = 4:26, Min = 6:31, Max = 25:27).

As a compensation, participants received individual feedback regarding their dietary behavior and nutrition knowledge based on their respective answers in the survey. This feedback was programmed and automatically viewed after the survey had been filled in completely. 

We used Unipark (Tivian XI GmbH, Cologne, Germany) as an online survey tool to collect data between May 2021 and November 2021. The study was conducted according to the guidelines of the Declaration of Helsinki. The Ethics Committee of the Medical Faculty of the University of Duisburg-Essen approved the conduction of the study (approval number 20-9718-BO on 26 November 2020, amendment accepted on 17 May 2021).

### 2.3. Study Variables

It was our objective to validate the GDBI-E in an English-speaking convenience sample, to verify construct and criterion validity. By verifying convergent and discriminant validity, construct validity can be ensured. Both convergent and discriminant validity make assumptions about correlations with similar and different constructs. Convergent validity assumes that similar constructs are significantly correlated, whereas discriminant validity assumes that different constructs should be uncorrelated [22]. Campbell and Fiske [22] argued in their paper that convergent validities exist when there is a significant correlation that is sufficiently high and different from zero. Discriminant validity describes a low level of agreement between constructs with different content indicating zero correlation. Precise cut-off criteria have not been set by Campbell and Fiske [22]. Current research has discussed and shown different cut-off criteria when analyzing convergent validities [23]. However, these cut-off criteria are not universally applicable to every field of research, which is why we referred to the definition of Campbell and Fiske [22]. We assumed positive correlations between GDBI-E score and attitude towards healthy food (single item) [24] as well as nutrition knowledge [25] to verify convergent validity. Interpersonal trust [26] as well as a belief about a just world [27] were captured to verify discriminant validity because these constructs have no content-related overlap but capture daily behavioral aspects and values of individuals. Criterion validity was assured by looking at interrelations between the GDBI-E and BMI (assessed via self-reported body weight and body height), life satisfaction (single item) [28], and physical and mental health (both self-developed single items). Furthermore, sociodemographic variables (age, gender, marital status, educational degree, the continent someone lives on, and native language) were assessed. Additionally, participants were asked about their general diet, food intolerances, and physical activity (single item asking about the number of days in the last month on which at least 30 min of exercise were performed) [29].

### 2.4. Statistical Analysis

Data analyses were carried out using R [30] and RStudio [31] combined with multiple packages [32,33,34,35,36,37,38,39,40]. In a first step, inferential analyses (correlation analysis, *t*-tests, ANOVA) have been performed to validate the GDBI-E score considering a significance level of *p* < 0.05. In step 2, a k-means cluster analysis as an unsupervised machine learning algorithm was conducted to examine whether the GDBI-E differentiates between people who eat more or less healthy regarding the nutrition recommendations of the WHO and the DGE. We used the elbow method and the average silhouette width to specify the optimal number of clusters and the silhouette coefficient for cluster evaluation [41,42]. The resulting group assignment was used to carry out correlation analysis to further ensure construct and criterion validity. Missing data were handled by listwise deletion. Cronbach’s Alpha was used for evaluation of scale reliabilities of convergent, discriminant, and criterion validity constructs. As the GDBI-E is based on a formative approach, the calculation of Cronbach’s Alpha is not appropriate for this construct.

## 3. Results

### 3.1. Descriptive Statistics of the GDBI-E

Table 1 provides the item statistics as well as the response distribution for each of the GDBI-E items and descriptive statistics for the GDBI-E score.

### 3.2. Sample Description and Descriptive Statistics

Table 2 shows the composition of the study sample regarding the sociodemographic variables, general diet, food intolerances, continent of living and native language. Moreover, Table 2 presents the GDBI-E scores regarding each sociodemographic variable. Mean age was 36.12 (SD = 12.80). Participants performed exercises of at least 30 min on 14.6 days (SD = 9.2) on average within the last month. The mean BMI of the overall sample was 24.8 ± 5.9 kg/m^2^. The mean BMI of women was 24.5 ± 5.9 kg/m^2^. The mean BMI of men was 26.6 ± 5.3 kg/m^2^. 

### 3.3. Inferential Statistical Analyses of GDBI-E Score and Sociodemographic Variables

Older participants had a higher GDBI-E score (*r* = 0.22, *p* < 0.001). Moreover, the GDBI-E score was significantly higher in participants with food intolerance than in participants without any food intolerance (*t*(245) = −2.15, *p* < 0.05). Further, participants differed in the GDBI-E score depending on their general diet (*F*(3,243) = 3.63, *p* < 0.05). Specifically, participants following a vegan diet had the highest GDBI-E score and participants following a vegetarian diet had the lowest GDBI-E score. The Tukey-HSD post hoc test revealed a significant difference for these two groups of general diet. However, the sample sizes between the general diet groups differed notably (see Table 2). Interpretation of the Tukey-HSD is limited because there was substantial variance in the relatively small vegetarian subsample (see Appendix A). Additionally, the GDBI-E score was positively related to the number of days on which exercises were performed (*r* = 0.29, *p* < 0.001).

The GDBI-E score did not differ between gender (*t*(241) = 1.26, *p* = 0.21), marital status (*F*(3,238) = 0.77, *p* = 0.51), educational degrees (*F*(6,235) = 0.48, *p* = 0.83), continents (*F*(5,237) = 0.42, *p* = 0.83), or native language (*t*(241) = −0.75, *p* = 0.46).

### 3.4. Construct and Criterion Validation: GDBI-E Score Intercorrelations

Reliability of the different scales regarding construct validation were 0.73 (nutrition knowledge), 0.82 (interpersonal trust), and 0.83 (belief about a just world). The scales reached acceptable to good reliability and were, therefore, included in the analysis. Table 3 displays the Pearson correlation coefficients of the GDBI-E score and the convergent, discriminant, and criterion validation scales. 

### 3.5. Construct and Criterion Validation: Cluster Analysis and Cluster Assignment Intercorrelations

Both the elbow method and the average silhouette width indicated that two clusters were most appropriate to conduct a k-means clustering. The mean silhouette coefficient for two clusters was *M* = 0.18. The clustering algorithm resulted in two distinct clusters with *n* = 77 for cluster 1 (mainly less healthy dietary behavior in correspondence to the international nutrition recommendations) and *n* = 186 for cluster 2 (mainly healthier dietary behavior in correspondence to the international nutrition recommendations). Figure 1 illustrates the partitioning clustering plot of the two clusters. Figure 2 depicts the item distribution of the 16 GDBI-E items within the two clusters on a z-standardized scale.

After assignment of the respective cluster to the dataset, we performed correlation analysis of the GDBI-E cluster with construct and criterion validity constructs. Table 4 shows the results of this analysis.

## 4. Discussion

The aim of this study was to validate the GDBI-E and to verify the construct and criterion validity of this instrument. Results of our correlation analysis confirmed the construct, as well as the criterion validity, for all scales but interpersonal trust. Convergent validity and criterion validity of the GDBI-E score could be fully confirmed. These results indicated that the GDBI-E was a valid instrument, which is associated with relevant similar constructs and was related to physical as well as psychological outcomes. Regarding discriminant validity, belief in a just world was unrelated to the GDBI-E score. Unexpectedly, interpersonal trust showed a significant interrelation with the GDBI-E score. Looking closely at the literature, former studies have shown associations of interpersonal problems and eating disorders in female samples [43,44]. In this context, attitudes towards other people are more likely characterized by distrust or uncertainty leading to avoidance of interpersonal situations. These attitudes or behaviors could be associated with negative affects or depressive symptoms triggering disordered eating [45,46]. This could be an explanation for our found correlation between interpersonal trust and dietary behavior. Beyond that, this result could be a statistical artefact related to our female-dominated sample.

Our study revealed the unexpected results that participants following a vegan diet had significantly higher GDBI-E scores than participants following a vegetarian diet. However, these results must be interpreted with caution because the Tukey-HSD post hoc test, and other possible post hoc tests, are sensitive to different variances in relation with imbalanced sample sizes in the sub-groups [47]. In the first study on the development of the GDBI, no differences between different forms of diet had been found [7]. 

With respect to the results of the cluster analysis, construct and criterion validity of the GDBI-E could be further confirmed. Correlations of the GDBI-E cluster assignment and the criterion validation constructs confirmed the score-related results and showed that participants with healthier dietary behavior regarding the nutrition recommendations of the WHO and the DGE were more likely to have a lower BMI and to perceive higher life satisfaction and health status. Moreover, regarding the convergent and discriminant validity, the GDBI-E cluster assignment was related to the respective constructs as theoretically expected. The cluster distribution of the items (Figure 2) revealed that the two clusters differed clearly in their dietary behavior regarding the compliance to the nutrition recommendations of the WHO and the DGE. However, participants with healthier dietary behavior differed from the nutrition recommendations in GDBI-E item 2, which indicated the intake of animal products. This possibly indicates that people with healthier dietary behavior deviate from the nutrition recommendations on this point and eat less animal products as recommended.

Generally, the nutrition recommendations of WHO and DGE are well established in Western civilizations and are used to evaluate dietary behavior [16,20,48]. However, looking at current definitions of a healthy diet, it becomes evident that the recommendations of the WHO and DGE do not reflect the latest state of research in accordance with a sustainable and diverse diet (e.g., [49]). Fresán and Sabaté [49] recommended a vegetarian diet to reduce the use of natural resources and to have a healthier lifestyle. Further, the planetary health diet represents a sustainable and healthy diet and has been evaluated and discussed in different research articles [50,51]. These recommendations focused on the interaction of health and sustainability of ecological habitats. However, these interactions are not represented in the recommendations of the WHO and DGE. These recommendations do not inform consumers about their influence on the environment with their diet. However, sustainable food cycles should be implemented to ensure a global supply of high-quality food products. Cacau and colleagues [52] showed that adherence to the planetary health diet is associated with lower BMI and waist circumference. Future studies should take novel nutrition recommendations into account, not only to consider individual health outcomes but also to have a look at public health and global health. Further, to reveal casual relations between the measured constructs longitudinal study designs should be conducted because dietary behavior is not a constant behavior over time and is influenced by individual aspects or external circumstances (e.g., emotional state, changing food preferences, or, currently, the COVID-19 pandemic) [53,54,55]. Furthermore, the BMI as a physical measurement has been a subject of criticism; following studies should also take other physical measurements, for example waist circumference, into consideration [56,57].

### 4.1. Limitations

Results of the study must be critically examined. Thus, limitations should be considered in this discussion. One limitation was referring to a selection bias. As participants were recruited via online resources, the study results are not generalizable on the overall population. Additionally, only 14% men took part in the study, indicating an overrepresentation of women. However, in gender-comparing analyses, no differences in GDBI-E scores between men and women have been found. Another sample-related limitation is that vegan participants were overrepresented in the study sample, reducing the generalizability of the results. Future studies should take this into account by collecting representative samples for gender as well as general diet. Limitations are also referred to methodological impairments of the GDBI-E. Only participants who completed all GDBI-E items could be considered because every item represented a specific dietary behavior and could not be substituted or replaced by other items of the questionnaire. Another methodological limitation was the reliability of the GDBI-E. Cronbach’s Alpha of the GDBI-E was not applicable. As dietary behavior reflects very different, uncorrelated behaviors, examination of reliability was inconclusive. Thus, the GDBI-E could only be evaluated if all items were completed. Moreover, the survey was conducted in a cross-sectional study design. Thus, the correlational results did not reveal causal relations between the constructs.

### 4.2. Strengths

After looking at the limitations of our study, we want to highlight the important strengths of this study. First, we developed a behavior-related questionnaire for the English-speaking population based on the results on the initial development of the GDBI [7]. Thus, the GDBI-E closes the gap to assess dietary behavior as a behavior-related construct and is a valid and reliable instrument which is accessible to the medical, as well as scientific, community. Second, the study results supported the findings of the German validation study [7], indicating that the GDBI-E is a valid instrument ensuring construct and criterion validity in an English-speaking sample. Third, we provided an efficient questionnaire, which operationalized the reliable recommendations of WHO and DGE measuring the relevant aspects of dietary behavior going beyond mere food or nutrient intake. Thus, practitioners, as well as scientific researchers, have the opportunity to discuss concrete dietary behavior with participants and are able to make health-related recommendations for a better dietary behavior. Our questionnaire is in line with current programs and initiatives emphasizing a healthy dietary behavior or disease prevention [58,59]. Fourth, our study showed that the GDBI-E was associated with physical measurements (BMI) as well as with psychological and physical perceptions of one’s own health status. These results indicated that the GDBI-E reflected different dimensions of health status and did not only focus on physical or psychological outcomes. Nevertheless, looking only at the BMI is not sufficient to make a statement about one’s health status because the BMI is not a proxy of body composition [60] and does not reflect mental or physical perceptions of oneself. Fifth, cluster analysis as a statistical method of unsupervised machine learning confirmed that the GDBI-E differentiated between two different dietary behavior clusters, resulting in a mainly healthy and a mainly less healthy dietary behavior in correspondence to the international nutrition recommendations.

## 5. Conclusions

In this study, we developed a new questionnaire to assess dietary behavior regarding the development of the German version of the GDBI. The instrument provides the opportunity to effectively assess dietary behavior in the English-speaking population based on international nutrition recommendation. The findings suggest that the GDBI-E is a valid instrument to assess dietary behavior in the English-speaking population. The construct and criterion validity of this new instrument could be confirmed and indicate adequate psychometric properties. Further, validity of the GDBI-E was confirmed by an unsupervised machine learning approach. Therefore, the GDBI-E is applicable to a broad population to assess dietary behavior based on international nutrition recommendations.

## Figures and Tables

**Figure 1 ijerph-19-02883-f001:**
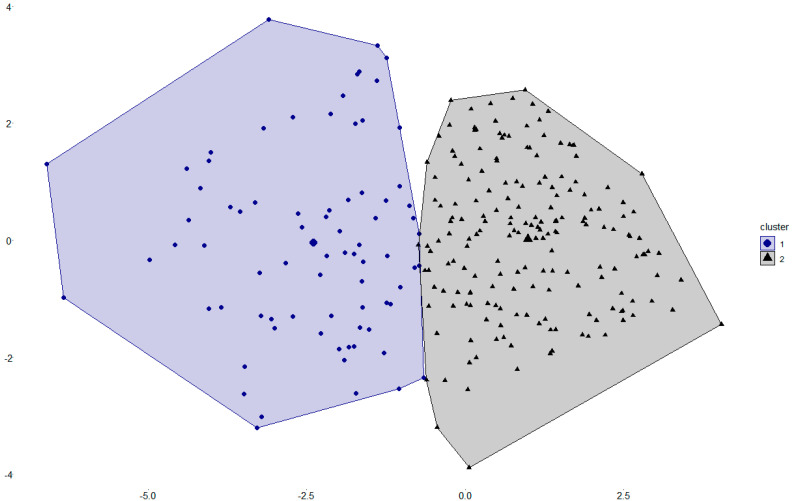
Partitioning clustering plot of the GDBI-E items. Cluster 1 represents participants who mainly follow a less healthy dietary behavior-related to the international nutrition recommendations. Cluster 2 represents participants who mainly follow a healthier dietary behavior-related to the international nutrition recommendations. The larger points indicate the respective cluster centers.

**Figure 2 ijerph-19-02883-f002:**
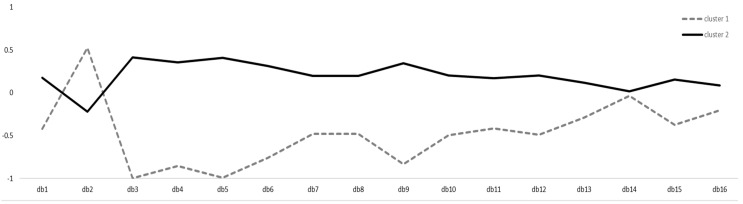
Cluster distribution over 16 GDBI-E items. The graph illustrates the distribution of the 16 GDBI-E items for the two clusters on a z-standardized scale. The two clusters represent a dichotomous scale (mainly less healthy and a mainly healthy dietary behavior).

**Table 1 ijerph-19-02883-t001:** Item statistics of GDBI-E items.

Item	Mean	SD	Skew	Response Distribution				
				1	2	3	4	5
db1	3.48	1.19	−0.30	3%	25%	14%	35%	22%
db2	2.68	1.61	0.29	38%	16%	9%	17%	21%
db3	4.43	0.97	−1.77	2%	6%	6%	19%	67%
db4	3.99	1.08	−0.80	2%	11%	17%	29%	42%
db5	4.20	0.91	−1.08	1%	5%	13%	35%	46%
db6	3.41	1.26	−0.32	8%	17%	24%	26%	24%
db7	3.93	1.21	−0.91	6%	8%	20%	22%	45%
db8	2.74	1.25	0.22	19%	28%	23%	20%	10%
db9	3.68	1.07	−0.54	3%	12%	23%	37%	25%
db10	3.93	1.23	−0.98	5%	13%	9%	29%	44%
db11	4.04	1.28	−1.19	7%	10%	8%	24%	52%
db12	2.72	0.95	−0.10	12%	23%	48%	14%	3%
db13	3.34	1.26	−0.24	9%	19%	25%	25%	22%
db14	3.70	1.46	−0.67	11%	16%	10%	17%	46%
db15	2.51	1.37	0.42	32%	24%	16%	18%	10%
db16	3.36	1.31	−0.28	10%	20%	20%	25%	25%
GDBI-E score	56.14	7.62	−0.46					

Within the possible GDBI-E score range of 16 to 80, the lowest individual score was 28 and the highest individual score was 75 in our sample.

**Table 2 ijerph-19-02883-t002:** Descriptive statistics of the study sample.

Variable	*n*	%	Missing *n* (%)	GDBI-E Score Mean (SD)
Gender			20 (8%)	
Female	206	78%		56.5 (7.4)
Male	37	14%		54.8 (7.8)
Marital status			21 (8%)	
Married	90	34%		56.0 (7.1)
Living in a relationship	70	27%		55.6 (8.3)
Single	74	28%		56.7 (6.8)
Other	8	3%		59.4 (9.3)
Educational degree			21 (8%)	
High school	18	7%		55.7 (7.2)
College	20	8%		54.3 (6.3)
Vocational training	13	5%		57.3 (6.8)
Bachelor’s degree or equivalent	103	39%		56.3 (7.4)
Master’s degree or equivalent	63	24%		56.7 (7.2)
Doctorate/PhD	20	8%		55.0 (10.3)
Other	5	2%		57.8 (7.7)
General diet			16 (6%)	
Omnivore diet	123	47%		56.1 (7.2)
Vegetarian diet	31	12%		53.1 (9.1)
Vegan diet	76	29%		58.1 (6.7)
Other	17	6%		54.8 (7.7)
Food intolerance			16 (6%)	
No	198	75%		55.7 (7.1)
Yes	49	19%		58.3 (8.5)
Continent			20 (8%)	
Africa	32	12%		55.5 (6.4)
Asia	13	5%		55.6 (7.5)
Australia and Oceania	35	13%		56.2 (7.5)
Europe	128	49%		56.6 (7.5)
North America	33	13%		55.5 (8.3)
South America	2	1%		62.0 (2.8)
Native language			20 (8%)	
English	171	65%		56.0 (7.3)
Other	72	27%		56.8 (7.8)

**Table 3 ijerph-19-02883-t003:** Pearson correlation coefficients for construct and criterion validity of the GDBI-E score.

Scales	GDBI-E Score
convergent validity	
attitude towards healthy food	0.33 ***
nutrition knowledge	0.19 **
discriminant validity	
interpersonal trust	0.13 *
belief about a just world	0.01
criterion validity	
BMI	−0.17 **
life satisfaction	0.15 *
physical health	0.27 ***
mental health	0.31 ***

* *p* < 0.05, ** *p* < 0.01, *** *p* < 0.001.

**Table 4 ijerph-19-02883-t004:** Pearson correlation coefficients for construct and criterion validity of the GDBI-E cluster assignment.

Scales	GDBI-E Cluster
convergent validity	
attitude towards healthy food	0.26 ***
nutrition knowledge	0.25 ***
discriminant validity	
interpersonal trust	0.10
belief about a just world	−0.06
criterion validity	
BMI	−0.15 *
life satisfaction	0.18 **
physical health	0.21 ***
mental health	0.25 ***

* *p* < 0.05, ** *p* < 0.01, *** *p* < 0.001.

## Data Availability

The data presented in this study are openly available in FigShare. Dataset: https://doi.org/10.6084/m9.figshare.17207432.v1 (accessed on 9 January 2022). R-Syntax: https://doi.org/10.6084/m9.figshare.17207408.v1 (accessed on 9 January 2022).

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
