# Peer review of "Validation of the English Version of the General Dietary Behavior Inventory (GDBI-E)"

_ijerph, 2022, doi:10.3390/ijerph19052883_

Round 1

Reviewer 1 Report

Current report demonstrated that General Dietary Behavior Inventory (GDBI)-E is applicable in research and clinical practice to assess dietary behavior in the English-speaking population. I like to give the following comments.

  1. GDBI has been proposed by same authors in 2021. Please add the application of GDBI by others to show the merits and specificity.
  2. German-speaking population is customed to English in general as shown in Table 1. Data from non-English speaking area will be more reliable.
  3. Variations of GDBI-E with GDBI remained unclear. Please describe it in the introduction section.
  4. Calculation of GDBI-E score is important. However, it was not indicated in clear.
  5. BMI is another essential factor in current analysis. It was also unclear.
  6. Variations between score and cluster shall be introduced in detail. Why the value of BMI was the same in both?
  7. Key points of the dietary behavior were not discussed in clear.
  8. Reliability of the questionnaire is another limitation that has been ignored.
  9. GDBI-E is associated with physical measurements (BMI) that needs more evidences.
  10. Authors developed a new questionnaire that was not shown in the context. Why?

Reviewer 2 Report

REVIEW

The article Validation of the English Version of the General Dietary Behavior Inventory (GDBI-E) presented for review was a validation of the existing General Dietary Behavior Inventory (GDBI), as the effective instrument, and verifying its construct and criterion validity. The purpose of the study was clearly outlined. The methodology and results chapters describe the correct course of the study and the obtained results. In my opinion, although manuscrypt has an average scientific value, it can be published in its current form.

Below are some remarks that may strengthen the manuscript

  1. Why were participants excluded who completed the questionnaire in less than 6:30 minutes (0.05 sample quantile) or more than 25:58 minutes (0.95 sample quantile)-what were the criteria?
  2. The research is not fully representative because the vast majority of the research were women - 86%, which should be taken into account in subsequent studies.
  3. Subsequent studies should include a relatively equal number of study participants representing different types of diets - so that they can be representative
  4. In the chapter research results, it would be important to provide information on how the individual eating habits were shaped in the study population

Reviewer 3 Report

1 – Pg 2, lines 96-97 - “exclusion criteria: As sum scores had to be calculated for dietary behavior and nutrition knowledge, participants who did not complete any items of these constructs were excluded. Also, participants who completed the survey in less than 6:30 minutes (.05 quantile of the sample) or more than 25:58 minutes (.95 quantile of the sample) were excluded. How was the “response time” criterion for the instrument created?

2 - Why did the authors analyze the scale from Cronbach's Alpha if the authors themselves state that such a procedure is not adequate?

3 - Were exploratory and confirmatory factor analyzes performed to determine the number of factors in the translated instrument and to identify whether this instrument behaved similarly to the original?

4 - Clarify whether the Cronbach's Alpha of 0.62 pointed out by the authors was from the construct and also, because it is not in the item: “Construct and criterion validation: GBDI-E score intercorrelations”? In this same item, the authors should add a reference about the acceptable values of Cronbach's Alpha.

5 - Add a table for the inferential statistics showing the differences between the variables (post-hoc results).

6 - When the authors say that: “The Tukey-HSD post-hoc-test revealed a significant difference for these two groups of general diet. However, the sample sizes between the general diet groups differed notably (see table 1)”, Why didn't they treat the data for different sample sizes?

7 - Add to the method item, the appropriate values for convergent validity and discriminant validity.

Round 2

Reviewer 1 Report

It has been improved in a good way.

This manuscript is a resubmission of an earlier submission. The following is a list of the peer review reports and author responses from that submission.